# Exploring barriers and facilitators in implementation fidelity of malaria screening intervention at Nepal-India border point-of-entry health desks-A mixed method study

Aney Rijal[1,2]*, E. Elsa Herdiana Murhandarwati[3,4], Megha Raj Banjara[5], Dilasha KC[6], Gokarna Dahal[7], Ari Probandari[8,9]

1 Faculty of Medicine, Public Health and Nursing, Universitas Gadjah Mada, Yogyakarta, Indonesia, 2 HERD International, Lalitpur, Nepal, 3 Center for Tropical Medicine, Universitas Gadjah Mada, Yogyakarta, Indonesia, 4 Department of Parasitology, Special Programme in Implementation Research, Universitas Gadjah Mada, Yogyakarta, Indonesia, 5 Central Department of Microbiology, Tribhuvan University, Kathmandu, Nepal, 6 Department of Public Health, Kathmandu School of Medical Sciences, Dhulikhel, Nepal, 7 Epidemiology and Disease Control Division, Department of Health Services, Kathmandu, Nepal, 8 Center for Tropical Medicine, Universitas Gadjah Mada, Yogyakarta, Indonesia, 9 Faculty of Medicine, Universitas Sebelas Maret, Surakarta, Jawa Tengah, Indonesia

* rijal.aney@gmail.com

## Abstract

### Purpose

This research aimed to identify the barriers and facilitators in implementation fidelity of malaria screening at Nepal-India border Point of Entry (POE) health desks.

### Method

A mixed-method approach was used guided by an implementation fidelity framework. Epidemiological records of reported malaria cases at selected border posts of Sudurpaschim and Lumbini province were obtained, while observation of the malaria screening was done among them. Qualitative inquiries were done with the health workers working at selected POEs, health coordinators and development partners working at respective districts, including representatives from three tiers of government: local municipalities, provincial and federal government. The suspected and confirmed migrant population were also interviewed about their experience while seeking for screening facility provided at POEs. Descriptive and trend analysis was done for number of tests per month and total malaria cases identified at POEs, while thematic analysis was performed for qualitative data.

### Results

There were fluctuating testing trends from March 2021 to December 2022, where malaria screening was peaked before and after rainy season and during festivities

**Data availability statement:** All relevant data are within the manuscript and its Supporting Information files, which also includes minimal data set.

**Funding:** The author(s) received no specific funding for this work.

**Competing interests:** The authors have declared that no competing interests exist.

and showed its the decreased priorities with the decreasing COVID-19 cases. Over those two consecutive years, 10 malaria cases were identified at POE in two provinces. Adherence to the screening protocol was partially followed, as there were limited guidance for malaria testing protocols at POE, incomplete fulfillment of health declaration forms, non-compliance in providing counseling during testing. Reported unavailability of test kits, variation in eligibility criteria, porosity of border, inadequate health workers, inaccessible infrastructures, limited awareness activities, fear of losing belongings among migrants, and administrative barriers were hindering the screening process. In contrast, presence of well-equipped infrastructures, perceived risk among the migrant's population and regular monitoring during the monthly review meetings at the local health institutions facilitated the health workers for conducting screening.

## Conclusion

The study concluded that prioritizing malaria detection and addressing barriers to ensure fidelity of malaria screening intervention is crucial.

## Introduction

Malaria is vector-borne disease characterized as an acute febrile illness caused by the bite of a female anopheles mosquitoes, infected by *Plasmodium* species [1]. It is endemic in 85 countries and areas, and remains a significant public health concern globally, with an estimated 249 million cases and 608,000 deaths in 2022 [2]. The WHO South-east Asia region estimates 5.2 million cases accounting for a 2% burden of global malaria cases. It is endemic in nine countries, disproportionately affecting vulnerable populations [2]. India alone accounts for higher percentage (65.7%) of total malaria cases estimated in the region [2,3].

Border malaria refers to the transmission of malaria that occurs across or along borders between countries sharing a land border [4]. It poses the challenge for elimination to the countries like Nepal which shares porous border with India, having high endemicity of malaria transmission. Nepal is one of the participating countries in E-2025 initiative to halt malaria transmission by 2025, while India is intensifying efforts on reducing the disease burden [5,6]. Although Nepal has considerably made progress by decreasing the number of positive malaria cases over the past five years committed for achieving the elimination. However, the proximity and porosity of border leading to unregulated movement of migrant population towards southern border of Nepal, resulting importation of malaria cases towards Nepal [7].

With the commitment from National Malaria Strategic Plan (2014–2025), to eliminate malaria cases in Nepal, there has been intensification of malaria testing at border posts, health facility and including active case detection at community levels [8]. National data showed that there has been increase in the positive malaria cases from 377 in fiscal year (FY) 2020/21–533 in FY 2022/23, where out of total malaria

cases, 95.6% were imported [9]. The majority of *P. falciparum* infections (97.36% of the total imported cases) showed an increasing in trend, rising from 13.53% to 28.33% between FY 2020/21–2022/23 respectively [9].

Malaria transmission is problematic as the majority of imported cases coming from the returnees or migrant workers arriving from the states of Maharashtra and Gujarat which are beyond the adjacent districts of India, while migrant workers returning from Middle East and African countries are also infected with malaria [9]. These regions significantly contribute to malaria transmission in Nepal, leading to higher likelihood of local foci transmission [7]. With the porous border provision with India, it pose severe concern for achieving the goals although cases have declined by 69% from 2009 (3500 cases) to 2018 (1065 cases) [8,10]. The migrant workers working at agricultural seasons are more prone to risk of malaria [7]. Consequently, the country would possibly miss the timeline set for achieving the target for elimination and ultimately leads to threat for resurgence, unless adequately addressed. The National Malaria Strategic Plan of Nepal emphasizes year-round malaria screening among migrants entering Nepal, necessitating enhanced screening strategies [8]. During COVID-19 pandemic, having shared risk of infections and common clinical signs and symptoms between COVID-19, HIV and TB and Malaria, integrated testing was conducted at 13 point of entry health desk of Nepal-India cross border [11,12]. Health workers comprising health assistants and laboratory assistants have been employed in the POEs [12]. The standard operating procedure (SOP) prepared by Epidemiology and Disease Control Division (EDCD) provides step-by-step guidance to the health workers in conducting integrated screening at POE. This involves taking history, screening, treatment and follow-up for the migrants [13].

Amidst the low-cost effectiveness of the intervention among others, it is important to understand how the protocols are being followed and what factors facilitates and hinder screening activities at point of entry [14]. Many countries aiming for malaria elimination face challenge related to border malaria and the lack of basic health services due to understaffing at POE [15]. This study aimed to provide the contextual factors influencing the implementation of malaria screening at border posts. The findings can help identify adaptive approaches to support adherence to protocols for other infectious diseases affected by cross border movement, particularly in Nepal. Additionally, the study has broader implications for malaria-endemic countries facing border malaria issues. Given Nepal's problem with imported malaria, this research could enhance its elimination strategies.

## Methods

### Study setting

The study was carried out in three PoE health desks of Kanchanpur, Kailali and Banke Districts of Sudurpaschim and Lumbini Province, Nepal as marked in Fig 1. These POEs has mandate for conducting integrated screening of COVID-19, malaria and tuberculosis, where Rapid Diagnostic Test (RDT) kits were used for testing malaria. These kits are procured through Management Division in coordination with Epidemiology and Disease Control Division (EDCD). These POE health desks were purposively selected because of high mobility of the migrant population from both Nepal and India side, and according to the malaria microstratification, 16 out of 22 high risk wards are located in these two provinces.

### Study design

The study employed a mixed-methods approach, which involved secondary data followed by observational methods, and qualitative inquiries. A concurrent mixed methods design facilitated multiple perspectives (i.e., federal government, provincial health directorate officials, health workers, police personnel, non-governmental organizations, and local municipalities)- enabling a comprehensive understanding of malaria screening implementation fidelity. Consolidated criteria for reporting qualitative research (COREQ) checklist is followed.

Implementation fidelity framework were used, proposed originally by Carroll et al., 2007 and modified by Hasson, 2010 [16,17]. Implementation fidelity is defined as the degree to which an intervention/program is implemented as it is proposed

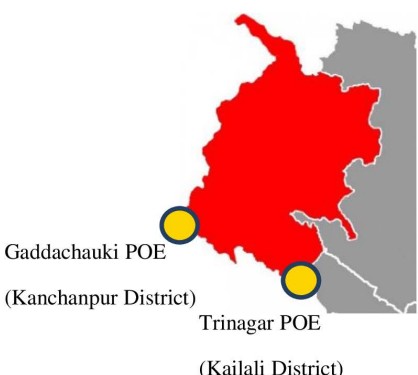
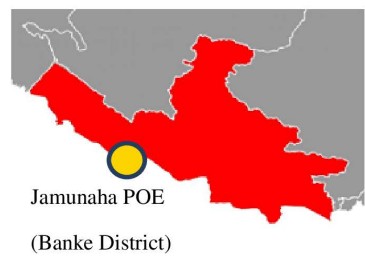

Gaddachauki POE

(Kanchanpur District)

Trinagar POE

(Kailali District)

**Sudurpashchim Province**

Jamunaha POE

(Banke District)

**Lumbini Province**

**Fig 1.  Selected point of entry health desks.**

or as recommended in the approved protocol. This framework consists of five major components to be measured: adherence, comprehensiveness of policy, strategies to facilitate implementation, quality of delivery and patient responsiveness. Program adherence refers to the degree to which program components are delivered as prescribed by the model and includes the subcategories of content, frequency, duration and coverage. Complexities of policy describe whether an intervention is vague or easy enough to be implemented or understood. Facilitation strategies are likely to optimize and standardize the fidelity. Participant responsiveness measures the degree to which extent the participants are accepting and enthusiastic about the interventions delivered by the providers.

### Data collection and analysis

Secondary data on total number of tests conducted per month and cases of malaria identified at border points were taken at two phases. Firstly, epidemiological records of three consecutive years (2020–22) were obtained from the national malaria program from Epidemiology and Disease Control Division (EDCD) of Nepal, particularly identifying the total malaria cases identified at all official POE, located at Sudurpaschim and Lumbini Province. Secondly, the data from malaria screening registers available at three selected health desks were collected since March 2021 until December 2022. The data was collected from March 2021, as the health workers officials conducted after the interim guideline for conducting integrated testing of COVID-19, Malaria and Tuberculosis. AR collected the total number of malaria testing conducted from March 2021 to December 2022, and it was calculated and visualized in chart format, while malaria cases were verified with the former collected data.

Non-participatory observations were conducted at three POEs through the use observation checklist, based on SOP and integrated screening literature [13]. Adherence to the screening protocol was observed while health workers are conducting malaria screening to the suspected migrant population.

Participants were chosen based on their availability during data collection. A guideline was prepared for each actor. Total of 26 key informant interviews (KII) were conducted with different stakeholders such as: health workers and development partners at three point of entry health desks (Gaddhachauki POE, Trinagar POE and Jamunaha POE), and administrative officials such as: adjacent local level municipality health coordinators, police officials, provincial health directorate officials and federal health officials. One FGD was conducted with health workers at Jamunaha POE. Total of 7 client interviews were taken with suspected and confirmed migrant populations at specific border posts. Total of 8 telephone inquiries were made with migrants who had tested positive for malaria in the

past six months, with a chosen timeframe to mitigate recall bias during interviews. The number of interviews varied based on reaching the saturation point, lasting between 15–30 minutes. Interviews were conducted anonymously, with informed consent obtained from all participants. Data was collected between 20-12-2022 to 3-02-2023 time period.

Quantitative data underwent descriptive analysis using Excel, visually presented through charts. Qualitative interviews, except for those with police officials, were recorded with written consent. Field notes complemented recordings for clarification and validation. Thematic analysis was performed using QDA Miner software, to identify barriers, facilitators, and adherence among health workers and migrant populations. Information triangulation with stakeholders ensured a comprehensive understanding.

### Ethical approval

Ethical approval was granted by the Medical and Health Research Ethics Committee of Universitas Gadjah Mada, Yogyakarta, Indonesia (Reg no. KE/FK/1517/EC/2022) and Ethical Review Board of the Nepal Health Research Council in Nepal (Reg no. 1294). Written consent was obtained from the participants. Anonymity and confidentiality of the individual participants were maintained.

## Results

### Characteristics of the study participants

Following Table 1 provides the characteristics of the study participants:

### Malaria screening trend in three selected borders

As shown in Fig 2, in the time interval between March 2021 to December 2021, 13117 migrants were tested at Gaddhachauki, 23449 migrants at Trinagar POE and 16598 migrants at Jamunaha POE. The screening trend has variation between these three POE. The screening trend remains consistent at Gaddhachauki POE, while Trinagar POE exhibited peaks in late February 2022 and September 2022. In contrast, Jamunaha POE experienced a peak from May 2021 to June 2021, followed by decreased testing, since late September 2022, largely due to festivals, elections, theft issues affecting the migrant population. The health workers were found to be reluctant for conducting testing as there was reduction of COVID-19 cases.

**Table 1. Characteristics of the study participants.**

| Character | Categories | KII | FGD | Client interview | Telephone inquiries |
|---|---|---|---|---|---|
| Gender | Male | 21 | 3 | 5 | 6 |
| | Female | 5 | 6 | 2 | 2 |
| Participants | Health worker | 9 | 9 | – | |
| | EDCD focal person | 2 | – | – | |
| | Local level municipality | 4 | – | – | |
| | Province representative | 2 | – | – | |
| | Health office | 3 | – | – | |
| | INGO | 3 | – | – | |
| | Migrants | | – | 7 | 8 |
| | Police officials | 3 | – | – | |
| Total | | 26 | 1 | 7 | 8 |

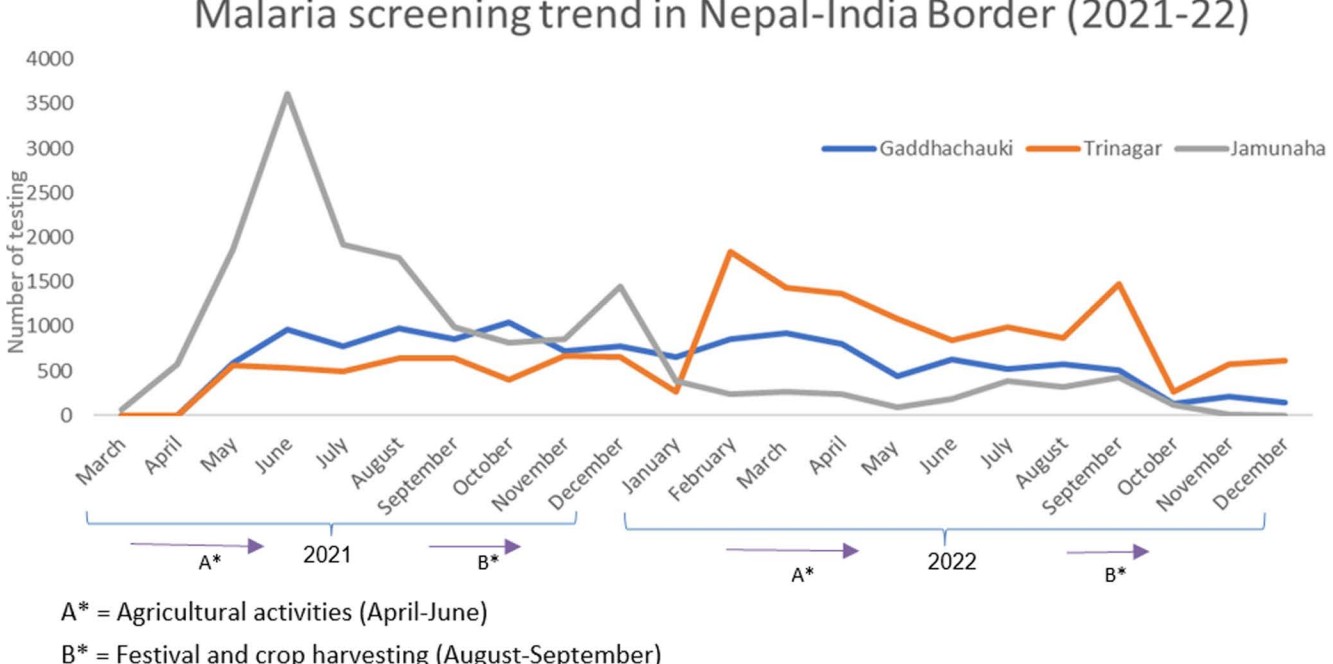

**Fig 2. Malaria screening trend in 3 selected borders (2021/22).**

The respondents reported that the malaria testing was intensified seasonally during the agricultural season and harvesting where focus was during the onset of rainy season (between month of June and August). The health workers reported that the testing rates often varies each day, and almost 40% of the migrant upon entering the Nepal side border.

A total of 10 malaria cases were identified between March 2021 and December 2022 in border posts located at Sudurpaschim and Lumbini Provinces, i.e., 5 cases per year. These cases identified were 8 males and 2 females, with ages ranging from 28.9±9.32 years. Most cases were of *P. vivax* (7 cases), followed by *P. falciparum* (2 cases) and 1 mixed species case.

## Adherence to the protocol

During observation of POEs, there were no hardcopy of SOP available at health desks, however, there was a poster to guide for testing malaria at some corners. The health workers faced challenge in fulfilling the health declaration form. The body temperature check was reported to be taken only when a person comes with fever, and it was not reported during observation. While conducting the tests the migrants were not well informed about the disease they were going to screen for and the time it would take to conduct the testing. It was found that the patient's names were not written on the cassette and identified based on the health workers' convenience. The health workers were not found to wait enough for 15–20 minutes to get the result from screening. The monitoring checklist were not filled properly.

Four themes were generated from the qualitative inquiries to explain the barriers and facilitators for malaria screening:

### Theme 1: Factors affecting the adherence to the protocol

a. **Reported unavailability of test kits**

The availability of test kits posed challenges to effective screening. In Gaddhachauki POE, there was a lack of malaria test kits, resulting in limited testing. It was reported that RDT kits were supplied through federal government, however there

was reported frequent stock out and delay in procuring them. In contrast, the federal officials mentioned that fewer tests were conducted at the point of entry due to fewer suspected cases, and not necessarily due to a lack of test kits.

### b. Differences in criteria used by health workers for targeting migrant population

The health workers used varied criteria for screening migrant population some health workers emphasized the critical role of obtaining travel history within 14 days for effective screening.

*"If they share to us that they came from that place or if we take their history, then it gets easier to conduct screening, because of which we have been able to find 6 cases from this border posts and are already recovered as well."* (KII, Health worker, POE)

Some health workers reported to conduct testing to only symptomatic cases staying for long duration at India, were tested for three diseases (malaria, COVID-19, and tuberculosis). Nepali citizens were primarily targeted for screening compared to Indian citizens, as the latter often had daily movement for shopping and were not sent inside the health desk by police officers. There were instances where Nepali individuals without belongings were suspected to have come from nearby places and escaped without testing. Pilgrimages were also exempt from testing at the health desk. In contrast, all migrants were tested upon the arrival, unless there was a shortage of the test kits.

*"The suspect did not meet today's criteria. Those people who have obvious signs and symptoms were tested. Also, the kits are less available…We are also not able to follow the protocol as standard as there would always be rush, we have to take care of many migrant population."* (KII, Health worker, POE)

### c. Porosity of the border

The porosity of the border allows the migrant population to enter and cross the border which leads to difficulty in controlling the movement of migrant population. Open borders and the movement of people make it difficult to identify individuals who pass through recently or have stayed in specific areas for a longer duration.
*"…there is a high burden of malaria cases in province due to the open border. However, not all points of entry have been approached, and testing are intensified based on the perceived risk." (KII, Provincial official)*

*"There were non-official borders like Bramadev, Tanakpur and Dodhara as a point of entry from where the migrant population enter, and screening were not done due to a lack of proper infrastructure and human resources."(KII_Healthworker_POE)*

### Theme 2: Structural and Service Factors

### a. Availability of health workers

The availability of health workers at the health desks posed challenges to effective screening. The desks were operated based on schedules assigned by the local municipality, which were opened based on the arrival time of the migrant population from the border. Police officials had 24-hour shifts, with four officials available during each shift to check migrants and send them for testing. Nevertheless, many migrants arrived at the border posts during evening, during closed hours of health resulting in missed opportunities for testing.

*"For both morning and evening shift, the health assistant and lab assistant must be there to ensure effective testing. As because one of them stay on leave frequently; it would be difficult to manage the flow of migrants."* (KII, Health worker, POE)

During interviews with migrants, the lack of available health workers at the health desk was cited as a reason for not undergoing screening.

*"…in the border, there were many people who come and go, while I came the health workers were not there to do a checkup, I didn't see anyone sitting there, and the police officials as well didn't stop me."* (client interview, migrant)

During the observation, the health workers were engaged in counseling to inform and convince migrants about the importance of malaria testing. They emphasize the benefits for migrants for knowing their health status and staying safe. Some migrants, however, reported for not being aware of the testing and have little knowledge about the diseases being tested. Police officials were also found to be participated in counseling and request migrants to facilitate the testing process.

*"…while returning from India, they noted the citizenship number and asked about Corona vaccine. We did not have the vaccine card. On the other side, there were 2-4 people in line, they were saying CORONA, CORONA, but on the other side people were allowed to leave, they were asking where you came from, we said from Bombay (Mumbai), then we had our belongings, and we came back."* (Client interview, migrant)

The recruitment process for permanent staff at POE was through public service examination, while temporary staffs were recruited through various levels, including the federal level, municipal level, provincial health directorate, NGOs, and INGOs. There were two staffs recruited from COVID-19 Response Mechanism (C-19 RM) specific to screen disease such as TB, HIV, malaria, and COVID-19. However, the stakeholders had concerned that recruitment of lower-level staff for planning and coordination showed incompetence and hampered internal collaboration with other authorities.

Meanwhile, there were inconsistencies in providing allowances to the staff members, particularly after the transition of leadership from the local level to the provincial level. Staff members responsible for screening malaria cases reported not receiving salaries until 6 months. The change in contract arrangement seemed to have caused a delay or disruption in the salary payment process, less motivated to work.

*"The migrants are going to the risky areas. Those health workers who work for 3 days got 3 months allowance but those who have been working every day at health desk wouldn't get allowance for even 3 days. Then the health worker would not be motivated, and the health facility wouldn't sustain."* (KII_POE_Health worker)

Health workers at the health desk demonstrated cultural sensitivity and a lack of discriminatory attitude towards the migrant population, which facilitated their testing. However, they find it challenging to deal with various behaviors from different people and stay motivated for their work. Additionally, health workers face security issues, including verbal abuse and aggression from the migrant population, which can be demoralizing.

*"It is a bit risky from other work, although there are risks at the hospital as well, but it is riskier than anything. People would come with HIV, TB, COVID-19. They would arrive from India and directly enter here. So, for that we shall need to take precautions."* (KII, Health worker, POE)

b. **Physical infrastructures and attribution of the screening area**

In terms of physical attributes of the screening area, each health desk was equipped with gender and disability-friendly toilets, a drinking water station, a breastfeeding corner, and a waiting post, supported by development partners, providing comfort to the migrant population. Gaddhachauki and Trinagar POE operated in temporary structures, while Jamunaha

POE had a semi-permanent structure. Gaddhachauki and Trinagar POEs had tunnels to prevent migrants from avoiding the health desk, which improved the effectiveness of screening. In contrast, there were some challenges related to the structures of POE. In Trinagar POE, health desk was operated in a temporary structure despite the availability of a permanent building. As there was reported lack of supervision in construction of the permanent building as per the standard construction design, leading to congestion and not meeting infection control standards.

*"…as our space is temporary, and it will be congested, and it would be difficult to manage and at other times there would be rainfall as well. We had pamphlets for malaria, but the monkeys would snatch and tear them. They would tear the tents, there would be theft of logistics and damage in lights and display board." (KII, health worker, POE)*

Regarding the location of the health desk, it was inconvenient for migrants, as the border posts were located far from the road network or marketplace and the exit area were close to the proximity of the temple, causing difficulty to navigate.

*"The land space is allocated through the sub-metropolitan office or customs office. As it is not constructed at the face of the point of entry, we are facing trouble to convince migrant population."* (KII, Provincial official)

c. **Limited awareness activities**

There were presence of posters and a few malaria-related pamphlets at POE, but the health workers were not found to disseminate them to the migrants as they were found to be busier in attending them in the crowded period. There was no targeted awareness program for migrant population, and police officials and transportation officials were the source of information about the screening for them after their arrival.

*"They would know only after they come here. They get information from police officials that they can leave only if they get tested otherwise, they will know only after they would come." (KII, health worker, POE)*

**Theme 3: Migrant population responsiveness**

There was a variation in behaviour and awareness among the migrant population regarding their need for screening. Migrant workers frequently experienced time constraints and a sense of urgency during their long journey back home. When health workers attempted individual screenings, it became challenging because migrants would feel tired, hungry, and rushed. The health workers reported that migrant population mostly have a fear of losing their belongings and security concerns.

*"…the migrants will arrive here with their belongings. They would go to the bus park by auto. The drivers will cross the border to India side and carry their stuff which it has been difficult for us. They would say they would keep their stuff and divert and elope." (KII, Health worker, POE)*

Health workers reported that integrated screening process takes time and they need to spend about half an hour at the health desk, including waiting for the results, which can make them annoyed and impatient.

*"…they are simple level labour workers. Those who are educated people, would cooperate, but some appears aggressively saying that they are in a rush and the testing is not necessary for them."* (KII, partner organization)

There was a lack of engagement to motivate migrants to retain for screening. It was reported that they would have burden and it costs them time and money. They have to leave their vehicles on the Indian side of the border, visit the health desk in Nepal, and then find another vehicle to continue their journey.

Generally, health workers viewed malaria testing at the POE positively, as many malaria cases can be imported from India however, some migrant population make excuses, claiming they already have a booster dose for COVID-19 and do not need testing for any disease. Also, they express concerns about not being informed about the screening process and whether they require to pay for the tests. In contrast, some migrants who were aware of the screening services, as informed by their colleagues, are more likely to get tested. Those who agree to the test have a perceived risk of malaria, especially if their place of residence was known to have a high mosquito population.

*"…malaria is high in Ahmedabad (Gujarat state) in India and even if there is fever it is malaria due to mosquito bite that is the disease of malaria."* (client interview, migrant)

Even though the testing process is hard, migrants expressed satisfaction and appreciated after knowing their status.

*"…testing is needed. The health workers stay here for 8-12 hours for whom? It is for us. We would not know that we might have some disease. We can also share with those we know. Earlier, it was not there."* (client interview, migrant)

## Theme 4: Contextual factors a. Monitoring and Evaluation

Health workers also reported irregular monitoring activities from health office, provincial health directorate and mostly limited to the online chat groups and occasional phone calls.

*"We do not visit PoE frequently. We have not been able to perform as per the schedule. Sometimes we visit twice in 3 months or once in 2 months. However, recently we had done a review meeting of POE during December, we had re-oriented the health staff as well."* (KII, Provincial official)

Some health workers expressed their concerns about limited knowledge and experiences due to limited opportunities for updating their knowledge. They reported not receiving any training or orientation even after working for several months. They also mentioned a lack of work appraisal, despite working 24/7 at the point of entry health desk, which shows negligence from the stakeholders.

*"I joined PoE in June 2022, it has been 6 months already, and I have not received any. It would be okay, if I had received as I got hired here about how we do data entry. Until now we have learnt by seeing others doing their duty."* (KII, Health worker, POE)

Monthly review meetings were held at the local level, involving health workers from the point of entry (PoE), the focal person from the province, and the health office. These meetings aimed to orient health professionals about malaria testing and treatment based on the guidelines of the Nepal Government. Stakeholders involved in the health desk, including the custom office, Chief District Office (CDO), and Maiti Nepal, receive comprehensive training on updated guidelines, policy strategies, and information about equipment, diagnostics, and treatment.

### b. Political Commitment and Administrative Barriers

Health workers reported facing administrative hurdles such as frequent transfers of security officials, interruptions during festivals and elections, and difficulties in managing migrants' belongings. They reported that the influence of police officials could make the migrant population obedient towards them and ensure screening coverage.

*"We must stay as guard to convince them to go inside the health desk, if they do not listen then there will be a fight in the crowd, we must shout, and we would not be able to convince them. The security personnel have uniform and power. We, the health workers cannot speak loudly"* (KII, Health worker, POE)

The leadership of health desks have shifted from the municipality to the provincial health directorate. Decision-making and resource allocation were centralized from the current FY, reducing the municipality's role in health coordination, financial support and assistance with equipment.

*"…health desk is located at our municipality, but our role in coordination, co-existence, and cooperation has diminished since the budget has been deducted and provided to the provincial health directorate."* (KII, Health Coordinator, Municipality)

Likewise, the stakeholders reported about the inadequacy of the budget for the malaria program, hindering regular orientation for organizations working at the border and security personnel. In the opposition, stakeholders emphasized the importance of conducting both community-based testing and border screening to identify malaria cases effectively.

*"…ultimately the patient goes to the community. if they are not screened in point of entry, our target must be community-based testing as ultimately people stay in the community, all disease is present in the community, our focus in terms of screening, firstly at point of entry and secondly at the community level."* (VBD inspector, EDCD)

There was limited support from the authorities of land vehicle and hotel services, creating barriers to the smooth functioning of health desks.

*"…the migrants will arrive here with their belongings. They would go to the bus park by auto. The drivers will cross the border to India side and carry their stuff. They would keep their stuff and elope the migrants."* (Police officials)"

c. **Adaptation in the integrated screening system**

The health workers reported that there were both challenges and opportunities during COVID-19 for testing malaria cases. They reported that during the higher morbidity of COVID-19, it was mandatory to conduct testing at the POE, which enabled to conduct malaria screening as well. However, by the time COVID-19 decreased, malaria was not taken much as a priority disease for screening. Decreased COVID-19 cases and the perception of lower transmission risk for malaria lead to reduced cooperation from migrants. The mandate to test only individuals without vaccine cards creates difficulties in counseling migrants for malaria testing.

It was found that if positive cases of covid-19 are identified, then they have to stay in holding centre in that way. It was reported that people take antipyretic 2–3 hours prior so that they won't have fever and won't have to be screened, if there is fever but COVID test is negative they must be kept in a holding centre and be tested and due to these things in malaria testing it has affected the actual testing people.

*"However, after the COVID-19 cases were reduced malaria was not taken much as priority like COVID and come for testing. Another thing malaria doesn't transmit from one person to other like COVID which can be transmitted while traveling in vehicle. That's why it is not taken as priority."* (KII, Health worker, POE)

Concerns about the reliability and sensitivity of Rapid Diagnostic Tests (RDTs) used for malaria screening also impact confidence in the test results and the prioritization of testing.

*"…what is negative part is the RDT that we used has cross reaction cross reaction means sometimes those who have scrub typhus this RDT has shown malaria positive those who have COVID positive also shows RDT positive our RDT validity is not enough […] due to that reason now our guideline says, if RDT is positive then it should be confirmed through microscopic and PCR […]. (KII, Provincial Official)*

Health workers reported that there were difficulties in reporting, as those hired for reported doesn't possess enough computer skills, despite their job titles as computer operators. There was presence of integrated management unit (IMU) for reporting COVID-19, Malaria and Tuberculosis, however, malaria is not reported into the system and reported unofficially through the online google system, which hinders effective reporting of line listings and total testing rates per day. Multiple information collection systems at the PoE create hassles for migrants, as they are asked for information repeatedly, leading to potential inaccuracies in reporting.

## Discussion

The study was done at POE health desks of Kanchanpur, Kailali and Banke districts where imported malaria cases persists. This study identifies the implementation fidelity issues based on the implementation fidelity framework, in terms of adhering to the protocol for conducting malaria screening, such as the inability to fulfill the health declaration forms, inadequate counseling provided to the migrants before testing, and not following the screening standards. In year 2022, only five cases were identified at POE from Lumbini and Sudurpaschim province. The reason for limited identification of the malaria cases in the borders and higher imported cases reported at the community level could be that the symptomatic cases are not easy to detect at border because of the porosity of the border. Likewise, various factors play into role such as unavailability of testing kits at POE. Health workers' adherence to collecting travel history and contact information supported migrants' health-seeking behavior. There were differences in using testing criteria, including exclusion of Indian citizens and pilgrims. Also, lack of awareness campaigns focusing on migrant population, structural factors such as far proximity of health desks from the entry point, and construction of the testing area not as per the standards, being inaccessible and leading to congestion and increasing the fear of losing belongings among migrant population. Irregular salary payments to health workers along with insufficient administrative support, including frequent transfers of police officials and security concerns contributed to the instability of the screening process.

This finding aligns with studies conducted at border entry points of Northern Ghana, Guinea, Liberia and Sierra, where few cases of Ebola Virus Disease (EVD) were detected despite high testing rates [18,19]. According to Cohen et al., there is a challenge in screening at land borders because of high mobility of migrants at formal and informal points [20]. Moreover, the World Health Organization (WHO) recommended that the government prioritize COVID-19 screening at point of entry for higher population mobility in seaports, airports and ground crossings [21]. However, the decrease in COVID-19 cases may lead to a decrease in political commitment at administrative level for point of entry screening. This may potentially overlook other diseases that may continue to pose public health risks. So, it is essential to prioritize other diseases of public health importance in integrated screening activities. Screening at the point of entry into and exit from countries is labour intensive, and the protective benefits associated with this type of preventive measures are contradictory, with limited public health impact, or evidence of success and benefits, of such measures [22].

As per the study, similar issues have been reported in other studies, emphasizing the need for stable political commitment, sufficient resource allocation and coordinated efforts to address administrative barriers effectively [23]. Integrated screening conducted at point of entry for multiple disease and equipped facilities illustrates facilitators for effective malaria screening. Improving infrastructure at POEs, including toilets and WASH stations, enhances migrant comfort and security as recommended by National regulations and International Health Regulations 2005 [24].

Challenges in detecting malaria among migrants, attributed to limitation of rapid diagnostic test kits. Studies suggests that endemic malaria, population movements, and foreign travel all contribute to the malaria diagnostic problems faced by the laboratory that may not have appropriate microscopy expertise available [25]. While alternatively there is a possibility that antibody tests could be possibly to be used by the health workers causing sub-optimal identification of the cases, which highlights the need for more reliable diagnostic tools. Ensuring the availability of antigen test kits is crucial. Once an antigen-positive result is obtained, it should be confirmed through microscopic examination to ensure accurate diagnosis and appropriate treatment [26]. This approach could mitigate the diagnostic problem.

Given that, there is a strategy to detected malaria cases a year-round at POEs, maintaining it is crucial to limit the local foci transmission to the high lands. So, it is essential to prioritize other disease of public health importance in integrated screening activities. There is the need for specifying the criteria to allocate resource more efficiently [7,27].

Finally, the study recommends following adoptive strategies that could be implied for strengthening the malaria screening at POE. Firstly, at health workers level proper training should be provided to them for adhering to the protocol and improving recording and reporting. There should be proper ways to increase their capacity of health workers to communicate with health coordinators and managers to establish collaboration between levels of governance, increasing the political commitment. At the system level, to increase the ownership of POE at local level, there should be reflection of activities at Annual work plan and budget, particularly for procurement of kits and logistics, upgrading the waiting area including whole physical infrastructures. Additionally, engagement activities like health camps and introducing audio-visual media can supplement screening efforts, highlighting the need for evidence-based guidance and sustainable facilitators to enhance screening coverage [28,29].

The strength is that the study employed mixed-method approach. It provides the quantitative evidence is supported by qualitative results through triangulation.

## Limitation of the study

This study has some limitations. Firstly, the representation was limited to only three out of thirteen health desks on Nepal-India border, as time constraints and resource limitations prevented the selection of all thirteen Points of Entry (POE). Secondly, it does not cover the perspective of migrants entering through informal entry points, which were not included in the study. Additionally, the study was unable to determine the numerical values required for testing, as it lacked denominators, i.e., data on the total number of migrants arrived by POE and the number of them tested.

## Conclusion

We found inconsistency in adhering to standard operating procedure, particularly in terms of incomplete fulfillment of travel forms and monitoring checklists. Challenges, including health desks located far from entry points, overcrowded screening areas, migrants' lack of knowledge, fear of losing belongings, and time constraints, contributed to dropouts from screening. Insufficient administrative support, irregular salary payments, and inadequate training for managing high migrant movements at entry points were human resource challenges. Conversely, factors that facilitated the screening process such as the perceived risk of exposure motivated migrants to participate and the dedication of health workers in challenging environments. Regular monthly reviews at the local level facilitated interaction with health workers and stakeholders. Despite the progress in reducing imported cases and halting malaria transmission at the point of entry health desks, challenges remain in ensuring consistent and high-quality implementation of malaria screening strategies. Role of cross-cutting or non-health actors plays the major role in ensuring the equitable and accessible screening facilities. Importance of local government in federal context plays the major role in ensuring the implementation strategies could enhance ownership. Addressing fidelity issues and leveraging these facilitators could enhance border malaria screening, could potentially prevent malaria transmission and safeguarding communities in countries sharing with border with imported malaria issues.

## Supporting information

**S1 File. Epidemiological records and interview guidelines** xxx.
(ZIP)

## Acknowledgments

We would like to express our gratitude to UNICEF/UNDP/World Bank/WHO Special Programme for Research and Training in Tropical Disease at the World Health Organization (TDR) and Universitas Gadjah Mada, Indonesia for providing postgraduate scholarship. We are grateful to the Epidemiology and Disease Control Division, Teku, Kathmandu, Nepal for

permitting us to conduct a study in point of entry health desk. We would like to extend our gratitude to Mr. Dinesh Koirala, Mr. Naresh Bikram Shah and Mr. Robin Maharjan representing development partners for supporting us during the research. Finally, we are thankful to all the participants who participated in the research.

## Author contributions

**Conceptualization:** Aney Rijal, E. Elsa Herdiana Murhandarwati, Megha Raj Banjara, Gokarna Dahal, Ari Probandari.

**Data curation:** Aney Rijal, Dilasha KC.

**Formal analysis:** Aney Rijal, E. Elsa Herdiana Murhandarwati, Megha Raj Banjara, Ari Probandari.

**Investigation:** Aney Rijal.

**Methodology:** Aney Rijal, Ari Probandari.

**Project administration:** Aney Rijal.

**Resources:** Aney Rijal.

**Software:** Aney Rijal.

**Supervision:** E. Elsa Herdiana Murhandarwati, Ari Probandari, Megha Raj Banjara.

**Validation:** Aney Rijal.

**Visualization:** Aney Rijal.

**Writing – original draft:** Aney Rijal, E. Elsa Herdiana Murhandarwati, Ari Probandari.

**Writing – review & editing:** Aney Rijal, E. Elsa Herdiana Murhandarwati, Megha Raj Banjara, Gokarna Dahal, Ari Probandari.

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
