## [Decision Letter · Decision Letter 0]

20 May 2024

PONE-D-24-09653Exploring barriers and facilitators in Implementation Fidelity of Malaria Screening Intervention at Nepal-India Border Point-of-entry Health DesksPLOS ONE

Dear Dr. Rijal,

Thank you for submitting your manuscript to PLOS ONE. After careful consideration, we feel that it has merit but does not fully meet PLOS ONE’s publication criteria as it currently stands. Therefore, we invite you to submit a revised version of the manuscript that addresses the points raised during the review process.

We look forward to receiving your revised manuscript.

Kind regards,

Edison Arwanire Mworozi, M.D

Academic Editor

PLOS ONE

Journal Requirements:

**Additional Editor Comments:**

Please address the reviewers comments and revise the paper!

Reviewers' comments:

Reviewer's Responses to Questions

**Comments to the Author**

1. Is the manuscript technically sound, and do the data support the conclusions?

Reviewer #1: Partly

Reviewer #2: Yes

Reviewer #3: Partly

2. Has the statistical analysis been performed appropriately and rigorously? 

Reviewer #1: No

Reviewer #2: Yes

Reviewer #3: Yes

3. Have the authors made all data underlying the findings in their manuscript fully available?

Reviewer #1: No

Reviewer #2: Yes

Reviewer #3: Yes

4. Is the manuscript presented in an intelligible fashion and written in standard English?

Reviewer #1: Yes

Reviewer #2: Yes

Reviewer #3: Yes

5. Review Comments to the Author

Reviewer #1: I would like to acknowledge all the authors of this paper for their hard work on this important issue. Saying that, I have few comments.

1. In the title, it is better to include the design of the study (mixed method)

2. Research gaps are not adequately addressed in the introduction part of the study.

3. In methods

- The study setting, it is better to say "the study is carried out" instead of "we carried out."

- Data collection and analysis, How do you maintain the quality of the data since you are using secondary data? For the qualitative part of your study, the researchers used key-informant interviews and FGDs, but the number of key-informant interviews and FGDs conducted in the study is not indicated. At last, I am not interested in the way quantitative data are analyzed.

4. The result part of the study need more work

Reviewer #2: COMMENTS

1: It would be helpful to include a sketch map showing the locations

2: The recommendation should be more specific, wherever possible it should be define to which authority they are targeted.

3: The abbreviation for point of entry (POE)should be defined in the Abstract and at first mention in the text and should be consistent ( POE or PoE)

4: Regarding concerns about RDTs. This is an important issue that should be further clarified: is there any technical confirmation of the true reliability of the tests? What measures are recommended to address this problem.

Minor edits:

64: risk=the risk

67: an= a standard

68: the points of entry

167: screening was not done

174: when a person

399: Study=The study

Reviewer #3: Dear Authors,

Your manuscript has a strong foundation, but there are some areas that could benefit from further refinement. Here are some specific comments:

1- Consider revising the language throughout the manuscript to ensure clear and impactful communication of your findings.

2- The abstract could be improved to more effectively convey the purpose and key takeaways of your research.

3- Some of the data, particularly regarding incidence and prevalence, may require updating with more recent information.

4- The positive number of cases is related to increased transmission or increase/improvement of testing?

5- Please ensure consistency in the fiscal year (FY) used when presenting data on P. falciparum infection and total malaria cases. When you talk about P.falciparum infection the baseline is 2019/20-2020/21 FY year, but when talk about the total number of malaria cases your FY is 2020/21-2021/22. please unify the information to have a better interpretation.

6- Please clarify how the data presented in lines 61-63 connects to the subsequent sections of the manuscript.

7- Is the novelty of your study specific to Nepal, or does it hold broader implications for other border regions?

8- Please introduce any abbreviations used in the text upon first mention (e.g., EDCD, IEC, CDO).

9- Please specify the method used to confirm positive malaria cases.

10- The data presented in lines 136-138 could benefit from further explanation to enhance reader understanding.

11- Clarify whether the "International Health Regulations 2005" are considered international or local regulations

12-Adding a section discussing the limitations of your study would further strengthen your work.

By addressing these points, you can significantly improve the overall clarity, strength, and impact of your manuscript.

6. PLOS authors have the option to publish the peer review history of their article (what does this mean? ). If published, this will include your full peer review and any attached files.

**Do you want your identity to be public for this peer review?** For information about this choice, including consent withdrawal, please see our Privacy Policy .

Reviewer #1: No

Reviewer #2: **Yes: ** Ahmed A Adeel

Reviewer #3: **Yes: ** Majid Asgari

---

## [Author Response · Author response to Decision Letter 1]

11 Oct 2024

Thank you very much for your comments. Response to the reviewer

Response: Okay, thank you.

Response: Okay, thank you.

Response: I would like to clarify that the funding provided is not in the form of a grant for research purposes, but rather as a postgraduate scholarship. Unfortunately, no specific funding has been allocated for research activities.

Response: Minimal data sets supporting the conclusions of this article are included within the article and its supplementary files. This includes the values used to build graph and the values used to share about the findings.

Reviewer #1: I would like to acknowledge all the authors of this paper for their hard work on this important issue. Saying that, I have few comments.

1. In the title, it is better to include the design of the study (mixed method)

Response: Inserted as suggested.

2. Research gaps are not adequately addressed in the introduction part of the study.

Response: The introduction includes the elaborated rationale now.

3. In methods

- The study setting, it is better to say "the study is carried out" instead of "we carried out."

Response: edited

4. Data collection and analysis, How do you maintain the quality of the data since you are using secondary data? For the qualitative part of your study, the researchers used key-informant interviews and FGDs, but the number of key-informant interviews and FGDs conducted in the study is not indicated. At last, I am not interested in the way quantitative data are analyzed.

Response: The epidemiological records of malaria is taken from government system (from Epidemiology and Disease Control Division) which is real time data and verified based on the case-based investigation. In terms of malaria testing data, it was obtained through reviewing the malaria register. The malaria register provides the information about the cases identified at POE, which was cross verified based on the line listing data provided by EDCD.

4. The result part of the study need more work

Reviewer #2: COMMENTS

1: It would be helpful to include a sketch map showing the locations

Response: Inserted

2: The recommendation should be more specific, wherever possible it should be define to which authority they are targeted.

Response: We have tried to included the specific recommendation by the end of the discussion.

3: The abbreviation for point of entry (POE)should be defined in the Abstract and at first mention in the text and should be consistent ( POE or PoE)

Response: edited

4: Regarding concerns about RDTs. This is an important issue that should be further clarified: is there any technical confirmation of the true reliability of the tests? What measures are recommended to address this problem.

Response: There is a possibility that antibody test were found to be procured/ or in used… that would be the reason, it is necessary to ensure antigen test kits should be ensured, and antigen positive microscopic test should be done for confirmation.

Minor edits:

64: risk=the risk

67: an= a standard

68: the points of entry

167: screening was not done

174: when a person

399: Study=The study

Response: Edited

Reviewer #3: Dear Authors,

Your manuscript has a strong foundation, but there are some areas that could benefit from further refinement. Here are some specific comments:

1- Consider revising the language throughout the manuscript to ensure clear and impactful communication of your findings.

Response: We have tried to revised, however, we still looking forward to the further suggestion if required.

2- The abstract could be improved to more effectively convey the purpose and key takeaways of your research.

Response: Edited and included based on the themes generated.

3- Some of the data, particularly regarding incidence and prevalence, may require updating with more recent information.

Response:

4- The positive number of cases is related to increased transmission or increase/improvement of testing?

Response: Nepal is currently in the elimination phase, and strategies for testing have been intensified at health facility levels, points of entry (POE), and within communities through active case-based testing. Consequently, we can assert that the testing capabilities have improved.

5- Please ensure consistency in the fiscal year (FY) used when presenting data on P. falciparum infection and total malaria cases. When you talk about P.falciparum infection the baseline is 2019/20-2020/21 FY year, but when talk about the total number of malaria cases your FY is 2020/21-2021/22. please unify the information to have a better interpretation.

Response: Edited, and added updated data.

6- Please clarify how the data presented in lines 61-63 connects to the subsequent sections of the manuscript.

Response: we have further elaborated on cases imported from India and other parts of world. Nepal shares a border with India, and the imported cases have been found not to originate from adjacent states of India, but rather from states such as: Maharashtra and Gujarat, which are of particular interest as border malaria burden further into achieving elimination.

7- Is the novelty of your study specific to Nepal, or does it hold broader implications for other border regions?

Response: The novelty of the study specific to Nepal has been further elaborated into the introduction section.

8- Please introduce any abbreviations used in the text upon first mention (e.g., EDCD, IEC, CDO).

Response: Edited.

9- Please specify the method used to confirm positive malaria cases.

Response: Secondary data were collected for the purpose of this study. This aim is not to conduct screening itself, but rather to analyse secondary data while concurrently engaging in qualitative inquiries with multiple stakeholders.

10- The data presented in lines 136-138 could benefit from further explanation to enhance reader understanding.

Response: Those data were not able to make a point, so was removed.

11- Clarify whether the "International Health Regulations 2005" are considered international or local regulations

Response: Both national and international regulations were considered and is elaborated in the discussion.

12-Adding a section discussing the limitations of your study would further strengthen your work.

By addressing these points, you can significantly improve the overall clarity, strength, and impact of your manuscript.

Response: The limitation section has been added.

---

## [Decision Letter · Decision Letter 1]

6 Dec 2024

PONE-D-24-09653R1Exploring barriers and facilitators in Implementation Fidelity of Malaria Screening Intervention at Nepal-India Border Point-of-entry Health Desks- A mixed methodPLOS ONE

Dear Dr.  Rijal,

Thank you for submitting your manuscript to PLOS ONE. After careful consideration, we feel that it has merit but does not fully meet PLOS ONE’s publication criteria as it currently stands. Therefore, we invite you to submit a revised version of the manuscript that addresses the points raised during the review process.

We look forward to receiving your revised manuscript.

Kind regards,

**Surya Bahadur Parajuli, MD**

Academic Editor

PLOS ONE

**Comments to the Author**

1. If the authors have adequately addressed your comments raised in a previous round of review and you feel that this manuscript is now acceptable for publication, you may indicate that here to bypass the “Comments to the Author” section, enter your conflict of interest statement in the “Confidential to Editor” section, and submit your "Accept" recommendation.

Reviewer #1: (No Response)

Reviewer #2: All comments have been addressed

Reviewer #3: All comments have been addressed

2. Is the manuscript technically sound, and do the data support the conclusions?

Reviewer #1: Yes

Reviewer #2: Yes

Reviewer #3: Yes

3. Has the statistical analysis been performed appropriately and rigorously? 

Reviewer #1: Yes

Reviewer #2: Yes

Reviewer #3: Yes

4. Have the authors made all data underlying the findings in their manuscript fully available?

Reviewer #1: Yes

Reviewer #2: (No Response)

Reviewer #3: Yes

5. Is the manuscript presented in an intelligible fashion and written in standard English?

Reviewer #1: Yes

Reviewer #2: Yes

Reviewer #3: Yes

6. Review Comments to the Author

Reviewer #1: (No Response)

Reviewer #2: (No Response)

Reviewer #3: Dear Authors,

Thank you for addressing my suggestions and questions. I appreciate the effort you put into revising the manuscript. However, I would like to raise some concerns regarding the method you used to present your revisions, which I found quite challenging to follow.

1. The use of the track changes feature has made the document overwhelming, as nearly all parts are highlighted in red, making it difficult to discern the updates. While I tried my best to interpret the changes, the process was not straightforward.

2. I did not receive a clear response regarding suggestion 3. Kindly provide feedback on this point in a more structured and transparent manner.

3. The phrase "any part of the world" is vague and requires clarification to ensure the statement is clear and meaningful.

4. Please follow standard formatting conventions by italicizing all species names used in the manuscript.

5. Include details about the suppliers of the kits used for detection in the methods section.

6. While the introduction has been updated, the references associated with this section have not been revised accordingly. 7. Please ensure consistency between the text and the references regarding the updated data for incidence and prevalence.

8. In your response, you mentioned the use of "secondary data." Could you elaborate on the meaning of this term and provide further context?

7. PLOS authors have the option to publish the peer review history of their article (what does this mean? ). If published, this will include your full peer review and any attached files.

**Do you want your identity to be public for this peer review?** For information about this choice, including consent withdrawal, please see our Privacy Policy .

Reviewer #1: **Yes: ** Adisu Asefa Kiyar

Reviewer #2: **Yes: ** Ahmed Adeel

Reviewer #3: **Yes: ** Majid Asgari

---

## [Author Response · Author response to Decision Letter 2]

8 Feb 2025

Response to the reviewer

1. The use of the track changes feature has made the document overwhelming, as nearly all parts are highlighted in red, making it difficult to discern the updates. While I tried my best to interpret the changes, the process was not straightforward.

Response: We appreciate your effort. We have attached both track change and clean version of the manuscript. This could bring some clarity on reviewing the changes.

2. I did not receive a clear response regarding suggestion 3. Kindly provide feedback on this point in a more structured and transparent manner.

Response: There is not availability of malaria incidence data for year 2024. So, we are mentioning data of fiscal year 2022/23.

3. The phrase "any part of the world" is vague and requires clarification to ensure the statement is clear and meaningful.

Response: This has been revised already.

4. Please follow standard formatting conventions by italicizing all species names used in the manuscript.

Response: Thank you. We have revised as per the standard format.

5. Include details about the suppliers of the kits used for detection in the methods section.

Response: Thank you. This sentence has been added in study setting, page no. 3, in line 111-112.

“These kits are procured through Management Division in coordination with Epidemiology and Disease Control Division (EDCD)”

6. While the introduction has been updated, the references associated with this section have not been revised accordingly.

Response: Thank you for pointing out. We have updated the references now.

7. Please ensure consistency between the text and the references regarding the updated data for incidence and prevalence.

Response: Thank you. We have cross-checked the references and revised.

8. In your response, you mentioned the use of "secondary data." Could you elaborate on the meaning of this term and provide further context?

Response: Thank you. We have clarified about secondary data in manuscript. In page no. 3, in line 152-152, under data collection and analysis.

“Secondary data on total number of tests conducted per month and cases of malaria identified at border points were taken at two phases.”

---

## [Decision Letter · Decision Letter 2]

3 Apr 2025

Exploring barriers and facilitators in Implementation Fidelity of Malaria Screening Intervention at Nepal-India Border Point-of-entry Health Desks- A Mixed Method Study

PONE-D-24-09653R2

Dear Dr. Rijal,

We’re pleased to inform you that your manuscript has been judged scientifically suitable for publication and will be formally accepted for publication once it meets all outstanding technical requirements.

Kind regards,

Khin Thet Wai, MBBS, MPH, MA

Academic Editor

PLOS ONE

Additional Editor Comments (optional):

Reviewers' comments:

Reviewer's Responses to Questions

**Comments to the Author**

1. If the authors have adequately addressed your comments raised in a previous round of review and you feel that this manuscript is now acceptable for publication, you may indicate that here to bypass the “Comments to the Author” section, enter your conflict of interest statement in the “Confidential to Editor” section, and submit your "Accept" recommendation.

Reviewer #1: All comments have been addressed

Reviewer #3: All comments have been addressed

2. Is the manuscript technically sound, and do the data support the conclusions?

Reviewer #1: Partly

Reviewer #3: Yes

3. Has the statistical analysis been performed appropriately and rigorously? 

Reviewer #1: Yes

Reviewer #3: Yes

4. Have the authors made all data underlying the findings in their manuscript fully available?

Reviewer #1: Yes

Reviewer #3: Yes

5. Is the manuscript presented in an intelligible fashion and written in standard English?

Reviewer #1: Yes

Reviewer #3: Yes

6. Review Comments to the Author

Reviewer #1: Generally, the authors have tried to address most of our previous concerns and comments.

Abstract: Better to start with background or introduction (not usual to start with the purpose)

Method: A mixed-method study was conducted. However, the result from the quantitative findings was not sufficient.

Recommendation: The authors have tried to identify several gaps in the qualitative interview results. However, the recommendations made by the authors were not sufficient enough.

Reviewer #3: Dear Authors,

Thank you for your responses; you have addressed most of my comments. I would like to kindly ask that if the kit used in your experiments is commercially available, please provide the details of the company that manufactures it, rather than the organization that supplied it to you. If the organization is the manufacturer of the kit, everything else seems fine.

7. PLOS authors have the option to publish the peer review history of their article (what does this mean? ). If published, this will include your full peer review and any attached files.

**Do you want your identity to be public for this peer review?** For information about this choice, including consent withdrawal, please see our Privacy Policy .

Reviewer #1: **Yes: ** Adisu Asefa

Reviewer #3: **Yes: ** Majid Asgari

---

## [Editor Report · Acceptance letter]

PONE-D-24-09653R2

PLOS ONE

Dear Dr. Rijal,

I'm pleased to inform you that your manuscript has been deemed suitable for publication in PLOS ONE. Congratulations! Your manuscript is now being handed over to our production team.

Kind regards,

on behalf of

Dr. PLOS Manuscript Reassignment

Staff Editor

PLOS ONE